# Angular Patterns of Nonlinear Emission in Dye Water Droplets Stimulated by a Femtosecond Laser Pulse for LiDAR Applications

**Yury E. Geints**

V.E. Zuev Institute of Atmospheric Optics, Zuev Square, 1, 634055 Tomsk, Russia; ygeints@iao.ru

**Abstract:** Femtosecond laser-induced fluorescence (FLIF) and femtosecond laser-induced optical breakdown spectroscopy (FIBS) are important tools for remote diagnostics of atmospheric aerosols using LiDAR (Light Identification Detection and Ranging) technology. They are based on light emission excitation in disperse media via multiphoton nonlinear processes in aerosol particles induced by high-power optical pulses. To date, the main challenge restraining the large-scale application of FLIF and FIBS in atmospheric studies is the lack of a valued theory of the stimulated light emission in liquid microparticles with a sufficiently broad range of sizes. In this paper, we fill this gap and present a theoretical model of dye water droplet emission under high intensity laser exposure that adequately simulates the processes of multiphoton excited fluorescence and optical breakdown plasma emission in microparticles and gives quantitative estimates of the angular and power characteristics of nonlinear emission. The model is based on the numerical solution to the inhomogeneous Helmholtz equations for stimulating (primary) and nonlinear (secondary) waves provided by the random nature of molecule emission in particles. We show that droplet fluorescence stimulated by multiphoton absorption generally becomes more intense with increasing particle size. Moreover, far-field plasma emission from liquid particles demonstrates a larger angular diversity when changing the droplet radius in comparison with multiphoton excited fluorescence, which is mainly due to the excitation of the internal optical field resonances in spherical particles.

**Keywords:** ultrashort laser pulse; water droplet; fluorescence; multiphoton absorption; plasma emission; angular diagram; remote diagnostics

## 1. Introduction

Femtosecond nonlinear laser-stimulated emission spectroscopy of disperse media, where the medium under study is a micron-sized particle which is luminous due to either laser-induced fluorescence (FLIF) of the active substance or because of the recombination of optical breakdown plasma (FIBS) generated in the particle volume, has recently received a new impetus in its development due to the use of femtosecond laser sources. The high intensity achieved in such ultrashort laser pulses leads to overcoming the energy threshold for the nonlinear optical processes of multiphoton laser absorption and optical breakdown of matter inside a particle and enables receiving reliable optical signals from emitting particles at considerable long ranges in the atmosphere using LiDAR technology [1]. In particular, fluorescence and FIBS LiDARs are created worldwide, which demonstrates their capability for determining the impurities in the atmosphere based on databases of fluorescence and emission species spectroscopy [2,3]. Nowadays, LiDARs are capable of measuring gas aerosol formation in the atmosphere at sufficiently long ranges (up to 3 km [4]) or extracting forest canopies from the urban landscape [5].

As known, at sufficiently high irradiation intensities far above the optical breakdown threshold $I_b$, laser-induced plasma can arise in aqueous medium [6], which produces spectrally resolved light emission of elemental plasma components as a result of radiative

recombination of generated free electrons with ions. For example, in an aqueous solution of table salt (sodium chloride), the spectral doublet of sodium with wavelengths near 590 nm possesses the highest emission intensity, which has been reported in many experiments on laser-induced femtosecond spectroscopy of model marine aerosols [7–14]. Importantly, the threshold for laser plasma formation in a spherical microparticle, in addition to the physical and chemical properties of particle substances, also depends on the parameters of the laser radiation (wavelength and pulse duration). According to published data [7,15,16], the threshold for optical breakdown in a water microdroplet exposed to a femtosecond pulse of a Ti:Sapphire laser ($\lambda \approx 800$ nm) is about $I_b \approx 10^{11}$ W/cm$^2$, which is two orders above the breakdown intensity of similar water droplets in air illuminated by a nanosecond laser pulse at the wavelength of the second harmonic of a Nd:YAG-laser ($\lambda = 532$ nm), i.e., $I_b \approx 2.5 \cdot 10^9$ W/cm$^2$ [17], and simultaneously two orders of magnitude lower than the breakdown threshold of pure air (without particles) when $I_b \approx 5 \cdot 10^{13}$ W/cm$^2$ [18].

If the incident laser radiation has a sub-threshold breakdown intensity, $I < I_b$, but a droplet contains a fluorescent dye, it is quite likely that laser-stimulated fluorescence can be excited. For example, for the greenish-yellow fluorescent dye uranine, the center wavelength of the wide spectral region of fluorescence lies near 525 nm, while the center of the absorption band is located near 500 nm. Obviously, since the photon energy in stimulating IR laser radiation is usually less than the energy gap between the absorption levels of a dye molecule, two-photon absorption fluorescence (TPA) through the main radiative transition of uranine will have the highest probability in such a spectral range. As a result, under TPA, a dye molecule absorbs two photons of incident radiation in a single quantum event (at the same time instant) and then releases its lower excited vibrational sublevel through a radiative transition to the ground singlet state.

Secondary radiation arising in a droplet volume (plasma emission or TPA fluorescence) is localized mainly in the vicinity of the internal focuses ("hot areas"), and upon exiting, the particle experiences multiple refractions and reflections on the spherical liquid–air interface. This causes the nonuniform character of the angular distribution of emission far from the particle (the receiver area). In addition, the particular shape of the droplet emission phase function is also influenced by the character of the emission mechanism, namely, the multiphoton-order (*m*) of the excitation process. As shown earlier [7,14,19], the higher the value of *m*, the more forward- backward-directed the angular distribution of nonlinear emission from spherical particles becomes. Moreover, light emission of sufficiently large water droplets (particle radius $a \gg \lambda$) is always characterized by an anomalously enhanced backward emission intensity [20].

In refs. [7,14], the angular characteristics of multiphoton-excited fluorescence in ethanol and methanol droplets with a radius of 25 to 40 μm were studied experimentally and theoretically. The droplets contain different dyes, coumarin 510 or tryptophan, and are irradiated by a Ti:Sapphire laser pulse with a duration of about 100 fs and an energy per pulse on the order of several microjoules. For one-, two- and three-photon excitations of fluorescence in microdroplets, central laser wavelengths of 400, 850 and 1200 nm were used, respectively. The main finding of these works is that the maximum fluorescence intensity of droplets is observed in the backward direction, i.e., towards the direction of incidence of the excitation pulse. Meanwhile, the ratio of fluorescence intensities in the backward and transverse directions increases when the order of multiphoton fluorescence excitation increases. A qualitative explanation of this effect was given based on the reciprocity principle of light rays emitted by fluorescence sources within a spherical particle. Numerical simulations established a relationship between the observed intensity of particle luminescence in the backward direction and the spatial localization of fluorescence sources, which increases when increasing the parameter m. Later, a similar effect was reported for non-spherical micro-objects, particularly in clusters formed by several micrometer polystyrene (PS) microparticles containing dry tryptophan [21].

In the meantime, the theoretical background for femtosecond LiDAR applications in cloudy environments is far from complete and requires further investigation. In particular,

this relates to the characteristics of droplet light emission stimulated by powerful laser radiation. In our previous work [22], the spatial location, effective volume and intensity of the excitation optical field in the "hot areas" of a micron spherical droplet were calculated within the Lorenz–Mie formalism. In particular, by the method of geometric optics, we show that the shape of the angular distribution of fluorescence excited in a spherical microparticle by laser radiation significantly depends on the morphology of the particle, and in particular on the specific location within the droplet and the effective power of the fluorescence sources. If the most intense of these sources is located near the shadow surface of the particle, the inelastic scattering phase function is elongated in the backward direction to the direction of laser pulse incidence. As the fluorescence source moves toward the particle center, the emission asymmetry disappears. In addition, the fluorescence emitted from the rear and front hemispheres of the droplet is characterized by different angular spreading. These results indicate that the size of aerosol particle and the specific type of nonlinear process causing the secondary emission are the key parameters affecting the angular structure of emission.

In present work, the development and dynamics of two-photon excited fluorescence and plasma emission inside an aqueous dyed spherical droplet are considered theoretically in particles exposed to high-power laser radiation. Based on the numerical solution of the Helmholtz equation by means of the finite element method, the angular structure of the secondary emission is simulated and studied in droplets with different sizes, allowing us to determine the directions of the maximal emission from the particles in the far-field.

## 2. Methods

Consider the following problem. A spherical droplet is placed in the air and illuminated by laser radiation with a wavelength $\lambda_0$ = 800 nm. Either saline water (NaCl solution) or a diluted aqueous solution of fluorescein (uranine, $C_{20}H_{12}O_5$) is considered as the liquid. The droplet is characterized by the radius a and the refractive index $n$ = 1.33 in the visible and near-IR spectral region (Figure 1a). In this spectral region, we neglect the linear light absorption of a liquid particle. As a result of optical wave diffraction incident on the dielectric sphere, regions with increased optical intensity are formed inside it, which usually are termed as the "hot areas" (HAs). These HAs offer favorable conditions for optical nonlinearity manifestation of a particulate substance and serve as the main sources of nonlinear light emission. Light emitted from HAs experiences constructive/destructive interference both inside and outside the particle provided by wave refraction and reflection at the boundaries. The resulting stationary emission field at blue-shifted wavelengths $\lambda_1$ (525 nm for TPA fluorescence and 590 nm for sodium plasma emission) is analyzed in terms of the angular intensity distribution in the far-field region.

From now on, we will make several simplifications to the problem under consideration. First, a stationary scattering problem is considered as we reduce the dimensionality of the original problem to only two spatial dimensions by assuming that all the optical fields (primary and secondary) possess azimuthal symmetry. This reduction neglects only the azimuthal field variation in the direction transverse to the direction of pump radiation incidence, but the angular emission structure along the polar angle θ is preserved.

The second problem simplification considers only monochromatic optical radiation for both the excitation optical field and the nonlinear emission from the droplet. This assumption is based on the fact that, even in the case of a high-power femtosecond pulse, upon its filamentation in air and supercontinual spectral broadening, the bulk of the optical energy is transported in a fairly narrow spectral range near the central wavelength [18]. Consequently, in the problem of droplet emission instead of a realistic wide-spectrum optical excitation, one can consider some artificial monochromatic radiation at certain effective wavelengths.

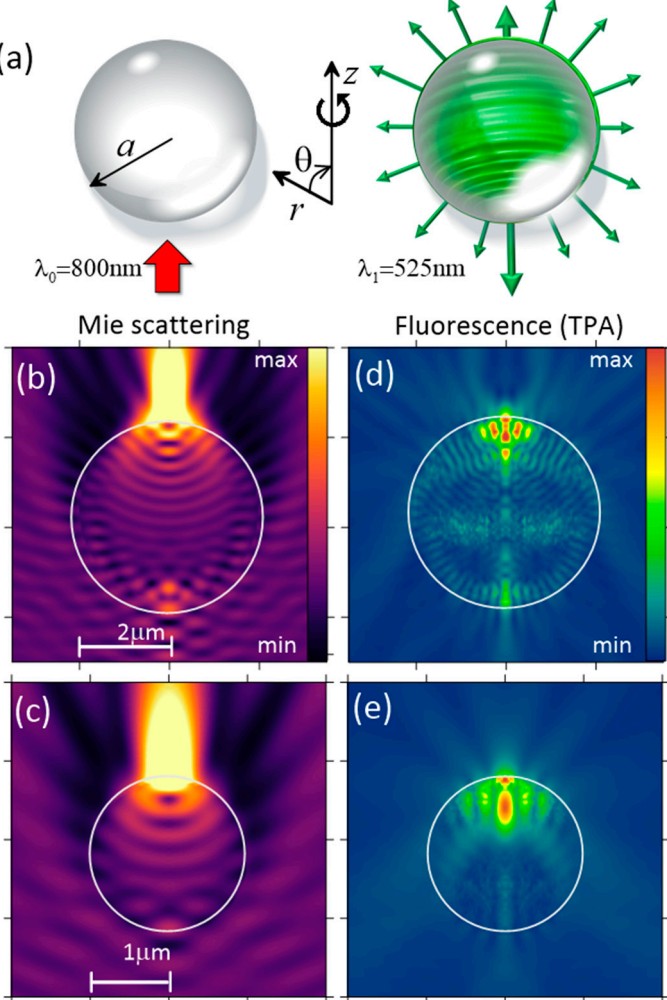

**Figure 1.** (**a**) Schematics of the problem statement on laser-stimulated nonlinear emission from a droplet. (**b**–**e**) Relative intensity distribution of (**b**,**c**) pumping radiation elastic scattering at $\lambda_0 = 800$ nm and (**d**,**e**) TPA fluorescence at $\lambda_1 = 525$ nm near water droplets with $a = 2$ μm (**b**,**d**) and 1 μm (**c**,**e**).

A similar approximation is adopted when modeling the nonlinear optical emission from the particle, i.e., the spatial dynamics of the optical field at only a single wavelength in the emission spectrum are considered. In the case of the breakdown plasma emission, this approximation corresponds to receiving the optical signal from only one selected chemical constituent of particle substance, e.g., sodium in a table salt solution. For the situation with multiphoton-excited fluorescence, which is characterized by a broad spectrum of dye luminescence, the spectral averaging of the working transition of the fluorophore molecule over energy sublevels as shown in [23] does not lead to critical errors in the measured emission signal if the fluorescing particles are of the mesowavelength scale ($a < 10\lambda$). Practically, this means the absence of the simultaneous excitation of several high-quality Mie resonances (the "whispering gallery" modes [24]) within the broadband spontaneous fluorescence spectrum of the microparticle, which can significantly modify the angular distribution of the emission. The requirement of the absence of strong resonances in the secondary emission allows one to neglect the modification of the quantum yield of spontaneous dipole emission excited by the resonant field, which is known as the Purcell effect [25].

Thus, within the framework of the approximations made, it is considered that each molecule of a particulate substance located at the point **r** inside a particle represents a dipole that first absorbs a certain portion of optical energy from the incident electromagnetic field $\mathbf{E}_i(\mathbf{r})$ at the wavelength $\lambda_0$ and then spontaneously (stochastically) emits an optical

quantum $\mathbf{E}_1(\mathbf{r})$ at a shifted wavelength $\lambda_1$. Without considering the nonradiative relaxation mechanisms, the rate $\gamma_d$ of spontaneous dipole emission (probability of dipole transition) is proportional to the scalar product of the dipole moment $\mathbf{p}_d$ of the corresponding energy transition and local field amplitude $\mathbf{E}_i$: $\gamma_d^{(m)} \propto \left| \mathbf{p}_d \cdot \left[ \mathbf{E}_i(\mathbf{r}) \right]^m \right|^2$, where $m$ is the number of simultaneously absorbed photons [26,27].

The governing equation describing the spontaneous emission from a spherical particle is the following inhomogeneous Helmholtz (wave) equation which is solved within the stationary problem formulation:

$$\left( k_0^2 \varepsilon_1 \right)^{-1} \nabla \times (\nabla \times \mathbf{E}_1(\mathbf{r})) - \mathbf{E}_1(\mathbf{r}) = \mathbf{P}_1 \{ \mathbf{E}_i(\mathbf{r}); \widetilde{\mathbf{r}}_1 \} \tag{1}$$

Here, $k_0$ is the wavenumber in free space, $\varepsilon_1 = n^2$ and the polarization source $\mathbf{P}_1$ depends randomly on the radiating dipole spatial position $\widetilde{\mathbf{r}}_1 = (\widetilde{r}, 0, \widetilde{z})$:

$$\mathbf{P}_1(\mathbf{r}; \widetilde{\mathbf{r}}_1) \propto \mathbf{n}_r I_i^{m-1} (\widetilde{\mathbf{n}}_d \cdot \mathbf{E}_i) \tag{2}$$

where we use the notation $I_i = |\mathbf{E}_i|^2$ for the optical intensity, $\mathbf{n}_r = \mathbf{r}/|\mathbf{r}|$ and $\widetilde{\mathbf{n}}_d \equiv \mathbf{n}_d(\widetilde{\mathbf{r}}_1)$ denotes a randomly oriented unit vector of the dipole ($\mathbf{p}_d = |\mathbf{p}_d| \mathbf{n}_d$). It follows from Equation (2) that the energy exchange rate between the optical field and radiating dipole in a unit volume, $dW/dt = (\mathbf{E}_i^* \cdot \mathbf{P}_1)$, in the case of, e.g., two-photon absorption ($m = 2$), depends on the squared intensity of the pump field as: $dW/dt \propto |E_i|^4 = I_i^2$.

Notably, when simulating plasma emission from a droplet, the volume density of secondary emission sources on the right-hand side of Equation (1) should be proportional to the free electron concentration $\rho_e$ (laser plasma is considered in equilibrium) generated through the photoionization of particle medium induced by the incident ("primary") radiation. The plasma polarization source reads as:

$$\mathbf{P}_1(\mathbf{r}; \widetilde{\mathbf{r}}_1) \propto \mathbf{n}_r \rho_e(\mathbf{r}) (\widetilde{\mathbf{n}}_d \cdot \mathbf{E}_i) \tag{3}$$

where the random character of dipole emission is also assumed.

In turn, the density of free electrons in the medium can be determined from the kinetic plasma equation accounting for the field (multiphoton) and impact (avalanche) medium ionization, as well as the decrease in the concentration of electrons due to the recombination with ions, as follows [6,28]:

$$\frac{\partial \rho_e(\mathbf{r}, t)}{\partial t} = W_I(\rho_{nt} - \rho_e(\mathbf{r}, t)) + \nu_c \rho_e I_i(\mathbf{r}, t) - \nu_r \rho_e^2(\mathbf{r}, t) \tag{4}$$

where $W_I$ is the photoionization rate (probability), $\rho_{nt}$ is the density of neutrals (molecules, atoms), $\nu_c$ is the avalanche ionization rate and $\nu_r$ is the electron recombination rate. In Equation (4), the diffusion of free electrons from the region occupied by plasma is not accounted for during the characteristic time of radiative recombination of femtosecond plasma [16]. Additionally, further considerations are made in the approximation of a short optical pulse when its duration is much less than the characteristic decay time of free electron plasma, which allows neglecting the term responsible for plasma relaxation in Equation (4).

Then, under the apparent condition $\rho_e << \rho_{nt}$ ($\rho_{nt} \approx 10^{22}$ cm$^{-3}$ is the critical density of free electrons when plasma begins resonantly absorbing optical energy), one obtains the following solution to Equation (4) for a rectangular pulse with the duration $t_p$ of stimulating laser radiation [28]:

$$\rho_e(\mathbf{r}, t_p) \simeq \eta_I I_i^{m-1}(\mathbf{r}) \exp(\nu_c I_i t_p) \tag{5}$$

Here, $\eta_I = \nu_I^{(m)} / \nu_c$, while $m$-photon water molecule photoionization with a certain rate $\nu_I^{(m)}$ is assumed, i.e., $W_I = \nu_I^{(m)} I_i^m / \rho_{nt}$.

By substituting (5) into (3), one obtains an expression for the plasma polarization source in a droplet, which is similar to Equation (2). In contrast to TPA fluorescence, the degree of multiphoton ionization of water molecules at a wavelength of 800 nm is considerably higher and is equal to $m = 5$ because bound electrons in a molecule need to overcome a high energy potential barrier of 6.5 eV to enter the conduction band and become free. In other words, not two but five photons of incident optical radiation are involved in a single excitation event of one plasma electron.

The exponential multiplier in (5) takes into account the "warming up" of free electrons by the optical field in a series of elastic collisions with heavy multicharged particles by the inverse Bremsstrahlung mechanism. This excess energy is used to increase the kinetic energy of a free electron and is converted into electron chaotic drift, which contributes to the development of the electron avalanche. The avalanche rate is proportional to the optical energy density ($I_i \cdot t_p$) at a selected point in the particle. For typical breakdown intensities of a femtosecond pulse, $I_i \sim 5 \cdot 10^{13}$ W/cm$^2$, with the duration $t_p = 100$ fs, by accounting for the data on $\nu_c$ in [16], one obtains the following estimate for the exponent index: $(\nu_c \cdot I_i \cdot t_p) \approx 10 >> 1$. This indicates that in the area of optical breakdown, the dynamics of the free electron concentration are not dependent on power, but rather experience an avalanche-like (exponential) growth with a linear increase in the pulse intensity.

Theoretical simulation of the nonlinear droplet emission is based on the numerical solution of Equation (1) by means of the finite element method implemented in the COMSOL Multiphysics software package. Since the emission of photons by a molecule/ion occurs spontaneously, in the numerical model, the emitting dipoles at each point of the particle are generally not correlated either in phase or in direction. Hereafter, in the steady-state approximation, we consider only the random direction of the dipole moments by specifying a random polar angle of the dipole $\theta \rightarrow \widetilde{\theta}(\mathbf{r})$ within the range $\widetilde{\theta} \in [-\pi, \pi]$ in the ($r$-$z$) plane. Then, at each point $\mathbf{r}$ inside the particle, the unit dipole moment vector can be represented as a function of a single random parameter: $\widetilde{\mathbf{n}}_d = \hat{\mathbf{r}} \sin \widetilde{\theta} + \hat{\mathbf{z}} \cos \widetilde{\theta}$, where $\hat{\mathbf{r}}, \hat{\mathbf{z}}$ are unit vectors along the corresponding coordinate axes. In the simulation, the statistics of droplet emission are gathered by setting different (usually a hundred variants) spatial distributions of dipole moments inside a particle with random dipole moment directions $\widetilde{\mathbf{n}}_d$. For each of these variants, problems (1)–(5) are solved separately. All the calculation series are eventually averaged over the spatial coordinates, and the resulting angular distribution of the nonlinear droplet emission is obtained in the far-field region ($r >> a$) by using Stratton–Chu integrals [29].

As known, this formalism is based on the assumption that the Green's function for the vector Helmholtz Equation (1) in the far-field region is known and the medium in this region is homogeneous. Then, the vectorial electric field of the emission $\mathbf{E}_{far}$ in the far-field region at any point $\mathbf{r} = (r,\theta)$ located on the surface of certain speculative sphere $S$ with radius $r$ and normal $\mathbf{n}$ which encompasses the particle is expressed through the integral of the near-field as follows:

$$\mathbf{E}_{far}(\mathbf{r}) = \frac{jk_0\mathbf{r}}{4\pi} \times \int_S [\mathbf{n} \times \mathbf{E}_1(\mathbf{r}_1) - \eta \cdot \mathbf{r} \times (\mathbf{n} \times \mathbf{H}_1(\mathbf{r}_1))] \exp(jk_0\mathbf{r}_1 \cdot \mathbf{r})dS \qquad (6)$$

Here, $\eta = \sqrt{\mu/\varepsilon}$ is the medium impedance and $j$ stays for $\sqrt{-1}$. Using Equation (6), one can calculate the angular distribution of the emission intensity $I(\theta) = |\mathbf{E}_{far}|^2$ of a water droplet by calculating the electromagnetic fields $\mathbf{E}_1$ and $\mathbf{H}_1$ directly near the particle surface.

## 3. Results

### 3.1. Polarization Sources of Droplet Emission

This section is devoted to the study of the spatial configuration of nonlinear polarization sources formed in a liquid microparticle by incident optical radiation and causing droplet emission in the anti-Stokes spectral wing. As an example, Figure 1b–e shows

2D distributions of the relative electric field intensity at the incident $I_i$ and shifted (TPA) $I_1 = |\mathbf{E}_1|^2$ wavelengths calculated for two radii of a spherical particle. Here, and hereafter, due to the model geometry used, the initial radiation is assumed to be circularly polarized relative to the incidence direction.

Analysis of the fundamental wave intensity $I_i$ profiles shows that inside an optically small particle with a radius $a\sim\lambda$, only a single HA is formed with an absolute maximum intensity located near the surface of the shadow hemisphere (Figure 1c). This HA serves as a source for TPA fluorescence providing the fluorescence distribution is also concentrated predominantly in the shadow part of the droplet (Figure 1e), although exhibiting a more complex spatial structure due to the interference of fields at multiple reflections from the inner particle rim.

In a larger droplet (Figure 1b), the intensity distribution at the incident wavelength is characterized already by two HAs located symmetrically in both hemispheres. At the same time, the intensity maximum in the shadow part of the drop still has a larger amplitude, which leads to the predominant concentration of the fluorescence also in the shadow droplet part. Additionally, besides the two main HAs located along the main sphere diameter, Figure 1d also shows the ring structure of the fluorescence extending along the entire particle surface, which is evidence of quasi-resonant excitation of an eigenmode (or several modes) in a spherical microresonator at shifted emission wavelengths [30]. As shown below, such situations cause sharp enhancement of the transverse-directed light emission of particles with certain resonant sizes.

The difference in the physical mechanisms causing the nonlinear polarizability of the droplet substance causes the different nature of the luminescence regions distributions. As seen in Figure 2a,b, inside a large water droplet (5 μm), the areas of maximal fluorescence excited by two-photon absorption ($m = 2$) are distributed along the sphere diameter with the front (illuminated) hemisphere fluorescing also quite strongly. Meanwhile, in the shadow hemisphere, one can see the luminous area of the "Descartes ring", the formation of which is typical during plane optical wave focusing by a transparent sphere [31].

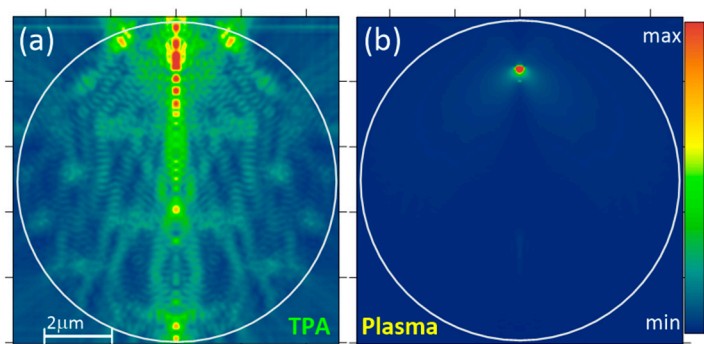

**Figure 2.** Relative intensity distribution of (**a**) TPA fluorescence (525 nm) and (**b**) plasma emission (590 nm) inside a water droplet with $a = 5$ μm.

Increasing the order $m$ of multiphoton absorption in case of particle matter ionization and plasma emission shown in Figure 2b leads to a drastic change in the polarization source configuration. Here, only one highly localized HA arises, located in the rear (shadow) hemisphere at a distance $r\sim0.8a$ from the particle center. In accordance with the reciprocity principle between the electromagnetic fields of the source and receiver [14], such a position of the nonlinear polarization maximum should lead to sharply pronounced backward directed plasma emission of the spherical particle.

Figure 3a–d summarizes the results of the HA parameters inside water droplets with different radii formed by optical radiation at the fundamental wavelength (800 nm). The HA parameters are obtained separately for the illuminated and shadow parts of a spherical particle.

The figures show the maximum intensity $I_m$, the rms-radius $R = \left[ \int\limits_S \left( \mathbf{r}^2 - z_m^2 \right) I_i dS / \int\limits_S I_i dS \right]^{1/2}$,

the $z$-coordinate of the maximum intensity $z_m$ for corresponding hot area and the effective absorbed in HA power $P_{\text{TPA}} = (I_m)^2 R^3$ via the two-photon excitation of the molecules.

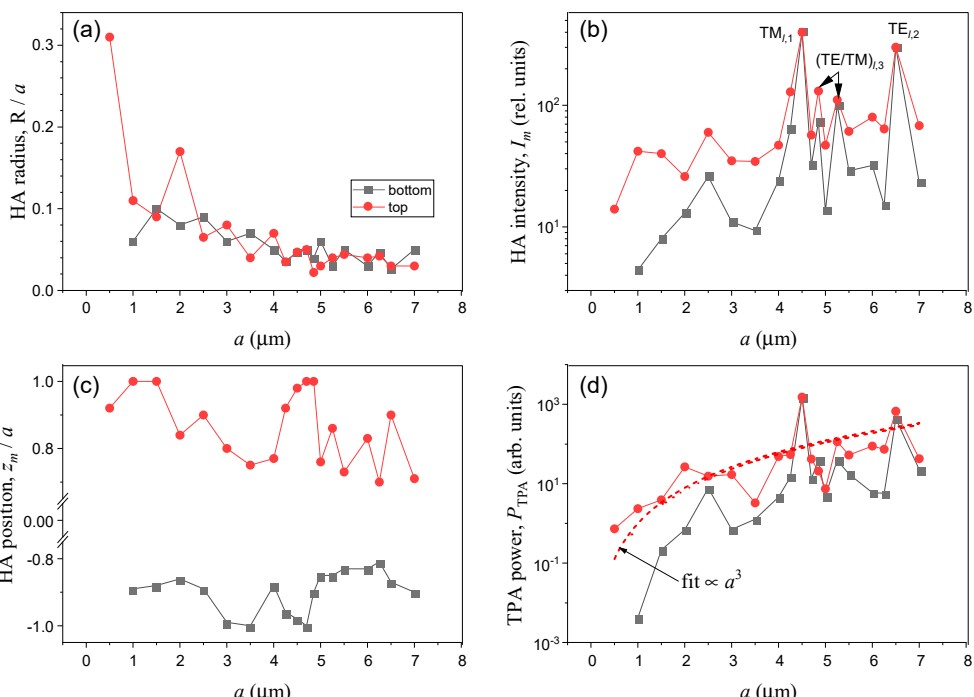

**Figure 3.** HA parameters ($\lambda$ = 800 nm) in water droplets of different radii. (**a**) rms-radius $R$, (**b**) maximal intensity $I_m$, (**c**) HA position $z_m$, (**d**) effective power of TPA source $P_{\text{TPA}}$. The parameter values in upper and lower hemispheres are shown by circles and squares, respectively.

From Figure 3a,c, it is clear that the relative size of the regions with dominating light absorption generally decreases monotonically with increasing particle size. Meanwhile, the HA position along the droplet diameter behaves in an oscillatory manner, but always staying in the range of $0.75 \leq |z_m/a| \leq 1$. The peak intensity $I_m$ of the optical field inside the HA (Figure 3b) demonstrates on average an increase with an increase in droplet radius, which can be explained by the increase in the collecting ability of a spherical optical lens (spherical droplet) when increasing its linear aperture (radius). At the same time, the quasi-monotone dependence $I_m(a)$ for certain particle radii is superposed with sharp intensity bursts. Analysis of the internal optical field structure shows that these bursts correspond to resonant excitation of particle eigenmodes, the field distribution of which follows that of a WGM. Some of these resonances are indicated in Figure 3b in "$(\text{TE/TM})_{l,p}$" notation, showing the state of wave polarization, as well as azimuthal ($l$) and radial ($p$) mode orders.

Generally, the dependence of the effective power for two-photon absorption $P_{\text{TPA}}$ (Figure 3d) on the droplet radius follows the proportionality $P_{\text{TPA}} \propto a^3$ with superimposed peaks coming from the internal field resonances. It is worth noting that in practice, when measuring the integral emission signal received from a polydisperse (in size) water aerosol, the relative contribution of the selected aerosol fraction to the received optical signal will be proportional not to the number of particles with the given sizes but to their volume fraction (water content).

### 3.2. Angular Structure of Droplet Emission (Phase Function)

Now we consider the simulation results of far-field laser-induced emission from water droplets. As mentioned above, the angular distribution of the particle emission intensity is calculated by integral expressions (6) obtained within the assumption that in the Fraunhofer

diffraction zone, i.e., far enough from a particle, only the scattered field exists, while the field of the incident wave completely vanishes. With this assumption, one can transit from a cylindrical to a polar coordinate system and consider the change in field vectors only in the polar angle θ. The phase functions $I(\theta)$ of uranine dye water droplets with different radii calculated for the elastic light scattering (Mie scattering) and nonlinear emission regimes are shown in Figure 4a–d. Recall that calculated data on the angular droplet emission presented below are statistically averaged over approximately one hundred variants of the random configuration of the emitting dipoles at each point of the particle. The standard deviation of the emission intensity for each angular direction depends on the nature of the simulated process and is about 30% for the TPA fluorescence and 12% for plasma emission.

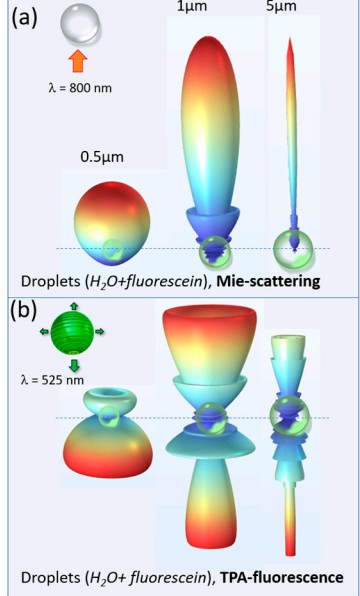
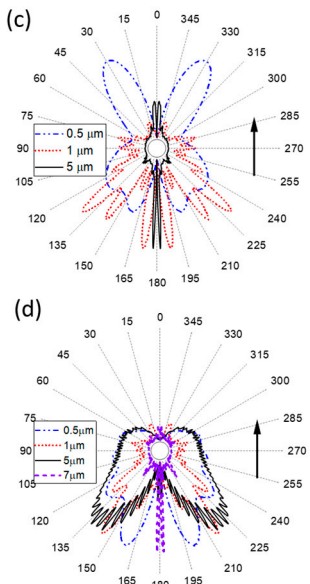

**Figure 4.** (**a–d**) Angular distribution of the far-field scattering intensity from dye water droplets of different radii. Three-dimensional diagram of (**a**) elastic Mie scattering and (**b**) two-photon fluorescence. (**c,d**) Intensity distribution of (**c**) TPA fluorescence and (**d**) plasma emission on the scattering angle θ (logarithmic scale). Arrows show the direction of incident excitation radiation.

The fundamental differences between the angular distribution of the linear scattering and the nonlinear two-photon excited fluorescence are clearly illustrated in Figure 4a,b. One can see that as the particle radius increases, the Mie scattering angular distribution evidently straightens in the forward direction (θ→0°), while on the contrary, the TPA angular distribution sharpens in the opposite direction, i.e., it has a maximum in the direction of the incident radiation (θ→180°). At the same time, in the transverse direction (θ→90°), both types of optical scattering show a decrease in intensity.

From a comparison of the angular structure of the fluorescence and plasma emission from micron-sized droplets shown in Figure 4c,d, it follows that the plasma radiation demonstrates a sharper backward directionality. Obviously, this is caused by the formation of only a single HA inside a spherical particle, which is located in particle shadow part and acts as a source of medium nonlinear polarization for the emission field. It is worthwhile noting that the TPA fluorescence of a droplet is almost always characterized by the presence of two polarization sources in both particle hemispheres.

## 4. Discussion

Consider an important quantitative characteristic of the angular diagram of the nonlinear particle emission, namely, the backward angular directivity $D$ of emission into the rear

(illuminated) hemisphere bounded by the polar angles $\pi/2 \leq \theta \leq \pi$, which is determined in the standard way:

$$D = 10 \cdot \log \left\{ \frac{\pi}{2} I(\theta_m) / \int_{\pi/2}^{\pi} I(\theta) d\theta \right\} \tag{7}$$

where $\theta_m$ is the angle at which the emission signal is maximal in amplitude, and the integration in the denominator (7) does not include the angle $\theta_m$. The parameters $\theta_m$ and $D$ as a function of particle size are presented in Figure 5a,b, respectively. Here, for comparison, the simulation results for three different physical processes of nonlinear droplet emission are shown, differing in the number of simultaneously absorbed photons.

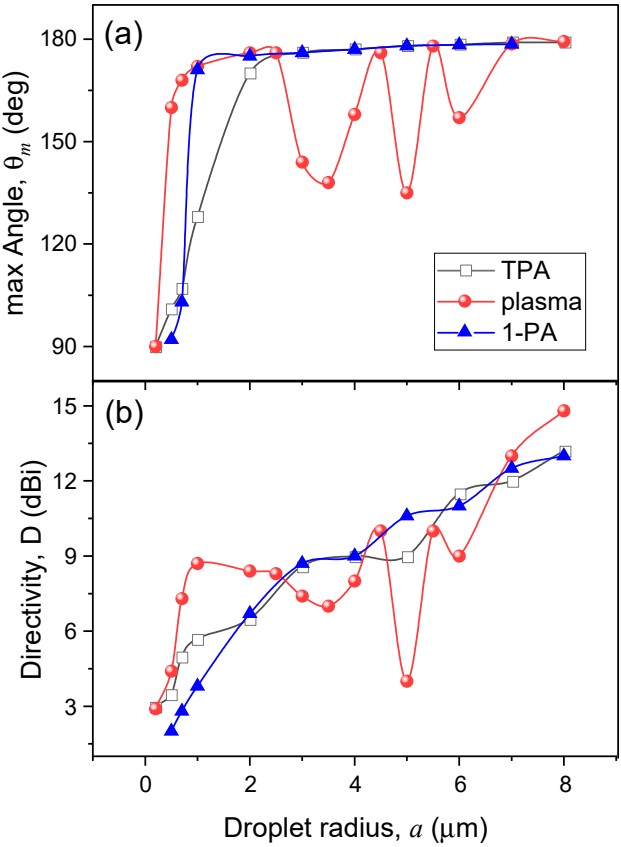

**Figure 5.** (**a**) Maximum emission angle $\theta_m$ and (**b**) backward angular directivity $D$ of water droplet emission with different radii resulting from the fluorescence of single- (1-PA) and two-photon (TPA) absorption, as well as from the optical breakdown emission (plasma).

As seen in Figure 5a, except for small droplet sizes $a \leq 1$ µm, where the emission has a dipole character ($\theta_m \approx 90°$), the emission maximum for all types of nonlinear scattering in general is observed at angles close to the backward direction, $\theta_m > 150°$. The change in the multiphoton order of dye fluorescence is affected the angle $\theta_m$ only for small- and medium-size particles, and this dependence is not monotonic when one transits from one to two photons and then to a sharper plasma absorption. Meanwhile, the angular emission directivity $D$ from the droplets (Figure 5b) normally increases with the increase in their size and is practically independent of $m$. One can conclude that, on average, the emission of medium-size and large water droplets has a pronounced angular orientation in the reversed direction, i.e., towards the source of primary excitation radiation.

Interestingly, the angular distribution of plasma emission from droplets exhibits a sharp nonmonotonic dependence on particle size that is not observed for TPA fluorescence. As can be seen in Figure 5a, the angle of maximum emission intensity demonstrates three

local minima when changing the drop radius from 1 μm to 6 μm. At these angles of minimal emission, a noticeable decrease in emission angular directivity is also realized. As our detailed analysis shows, this is caused by the specificity of the spatial position of the sources of nonlinear polarization in the particle.

Indeed, recalling Figure 3c, one can see that as the drop radius increases, there is a general tendency for the HA to move to the particle center. We carried out additional simulations of droplet emission when artificially setting the polarization (emitting) source in the form of a pointwise dipole, which physically corresponds to a single plasma region forming in the shadow part of a micron-sized drop such as that in Figure 2b. The angular dependence of droplet emission obtained in this case is shown in Figure 6. Evidently, as the dipole coordinate $z_m$ decreases, the angular distribution of the particle emission becomes more homogeneous, while the angle of emission maximum $\theta_m$ apparently decreases. This means that when the droplet size increases, the plasma emission originating from the optical breakdown inside the particle volume becomes more homogeneous.

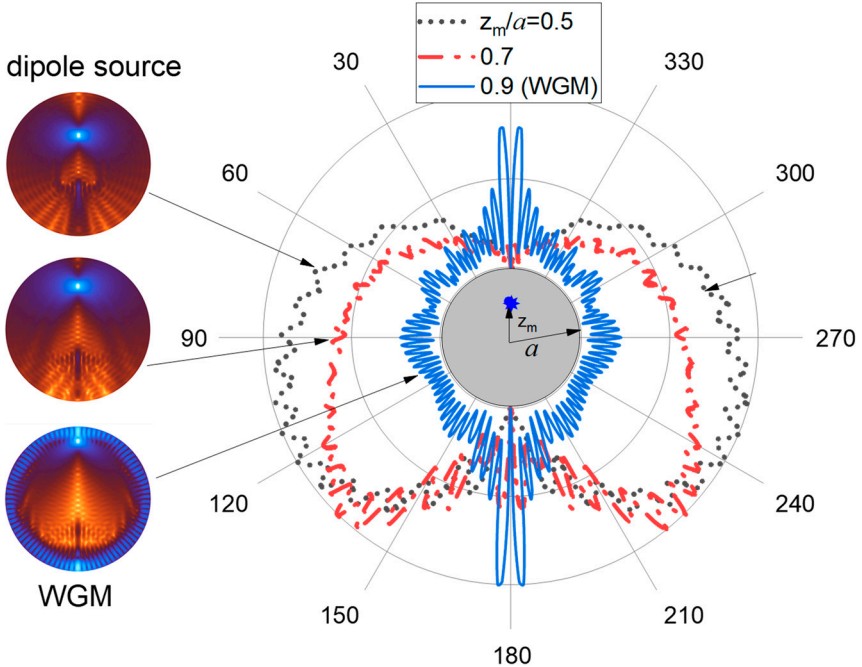

**Figure 6.** Model calculations of the angular emission distribution of a 5 μm droplet at different positions of the nonlinear polarization source represented by an electric dipole (left column).

At the same time, as noted above, for certain sizes of spherical particles there are possible situations when the internal optical field is drastically enhanced by the resonance WGM excitation. In this case, instead of a single point polarization source, one should consider two separated HAs located diametrically opposed near the surface of the spherical droplet. In Figure 6a, this situation corresponds to the case with $z_m/a = 0.9$. Under WGM resonance, the angular distribution of droplet emission becomes appreciably elongated and quasi-symmetric in the forward/backward directions and the value of maximal angle $\theta_m$ approaches 180°.

As seen in Figure 3b,c, the cases of excitation of the internal field resonances in the particle are observed near radii $a$ = 2.5, 4.5, and 6.5 μm. As a result, the oscillatory character of the dependence $\theta_m(a)$ for the plasma emission is caused by two opposing physical trends: (a) shifting the nonlinear polarization source closer to the particle center which leads to a $\theta_m$ decrease and (b) increasing the probability of WGM excitation inside a spherical particle, which causes an increase in the emission maximum angle. For more "smooth" processes of single- and two-photon excited fluorescence, such oscillatory behavior of the maximal emission angle $\theta_m$ with droplet radius is not observed because of the absence of such a

sharp disbalance of polarization sources in shadow and illuminated hemispheres during nonresonant excitation of the internal optical field.

As the key result of our studies, the normalized emission power $P_e$ of aqueous dye droplets of different radii is plotted in Figure 7. The values of $P_e$ are presented in arbitrary units for each type of nonlinear optical emission considered in the direction of maximal intensity $\theta_m$. For each type of nonlinearity, the normalization of $P_e$ values was performed in such a way that the different curves in the graph are clearly distinguishable. Importantly, a quantitative comparison of the results is possible only within each individual dependence.

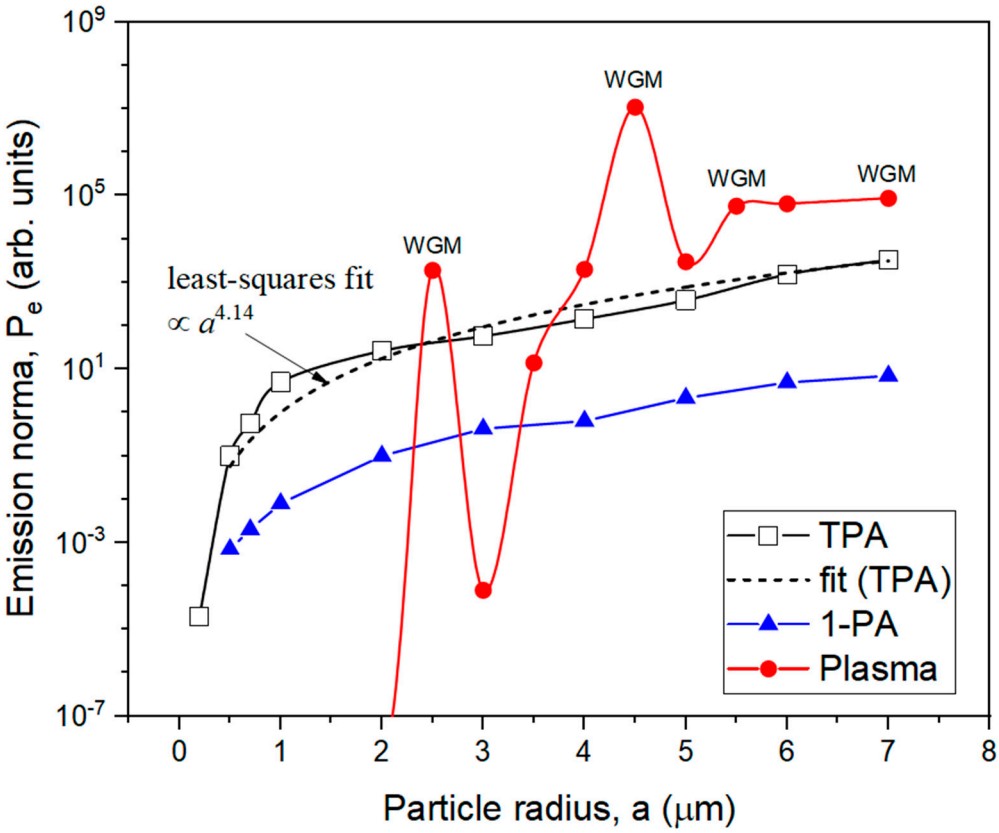

**Figure 7.** Relative emission luminance (power within $\theta_m$) of dye water droplets for different physical processes. Fitted dependence for TPA fluorescence is shown by dashed curve; the situations of WGM excitation in droplets are indicated for plasma emission.

Generally, an analysis of Figure 7 shows that the droplet emission via multiphoton absorption processes becomes more intense with increasing sizes of particles. Moreover, for the case of TPA fluorescence, this tendency can be approximated as $P_e \propto a^{4.14}$, which is steeper than the above-established increment for TPA polarization source power ($P_{TPA} \propto a^3$ in Figure 3d) and indicates better emission confinement within the angle $\theta_m$. A similar tendency for classical single-photon fluorescence yields a lower power degree, $P_e \propto a^{2.47}$. Additionally, in the case of plasma droplet fluorescence, this dependence becomes considerably sharper with a high influence of WGM resonance excitations and cannot be fitted by a power law.

## 5. Conclusions

In conclusion, the angular distribution of dye water droplet emission was theoretically examined when the droplets are exposed to high-power infrared laser radiation stimulating nonlinear optical processes of multiphoton absorption. Specifically, we consider dye fluorescence in two-photon absorption and emission of recombining optical breakdown plasma created inside the droplet by incident laser radiation. Using numerical simulations,

providing the random distribution of nonlinear polarization sources in the droplet volume, we calculate the angular diagram of emission in the far-field and obtain the dependence of emission angular directivity on the particle size and type of optical nonlinearity. It turns out that the droplet emission in multiphoton absorption generally becomes more intense with increasing droplet size. Meanwhile, backward directed droplet emission (toward the laser incidence) exhibits better angular directivity with increasing particle radius providing the maximum emission angle approaches the value of 180° (backward direction). Compared to two-photon excited fluorescence (TPA), the plasma emission of droplets demonstrates stronger angular variability with particle radius, which is due to the specific spatial position of the emitting plasma regions inside the spherical particle and the significant influence of eigenmode resonances.

We expect that the results of this work can be useful particularly for elaborating a theoretical model for the nonlinear emission of liquid aerosol particles stimulated by high-intensity, ultrashort laser radiation, which can extend the area of femtosecond LiDAR applications for remote diagnostics of atmospheric aerosols, including particle size and elemental composition analyses.

**Funding:** This research was funded by the Russian Science Foundation (RSF), grant number 21-12-00109.

**Informed Consent Statement:** Not applicable.

**Data Availability Statement:** Data available on request due to restrictions, e.g., privacy or ethical.

**Conflicts of Interest:** The author declares no conflict of interest.

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
