# Peer review of "Angular Patterns of Nonlinear Emission in Dye Water Droplets Stimulated by a Femtosecond Laser Pulse for LiDAR Applications"

_remotesensing, doi:10.3390/rs15164004_

Round 1

Reviewer 1 Report

In this study, a theoretical study on the angular models of nonlinear emission in dye water droplets excited by a femtosecond laser pulse was carried out. Here are my suggestions for improving the work:

1) Experimental proof of the results would have been more meaningful in terms of scientific contribution.

2) However, the literature search is limited. More publications on this subject should be examined.

3) It is also expected that the theoretical review will establish more relationships with LiDAR.

4) It is not understood what "Material" is in the Material and Methods section. This phrase can be removed from the title.

5) Discussion section should also be improved.

6) In the Conlusions section, suggestions for future studies on how the theoretical framework can be applied in the practical field should be added.

Minor editing of English language required

Author Response

Respond to Reviewer#1

We appreciate the time and effort of the reviewer for careful inspection of our work.

N.B. All the revisions are high lightened in green.

  1. Reviewer: Experimental proof of the results would have been more meaningful in terms of scientific contribution.

Authors: Thank you for this comment! Currently, the experimental verification of the theoretical simulation results presented in our paper is in the process. Specifically, we measure the angular distribution of Rhodamine-dye water aerosol fluorescence stimulated by powerful femtosecond pulses upon the filamentation in air. However, this is the topic of our future publication.

  1. Reviewer: However, the literature search is limited. More publications on this subject should be examined.

Authors: Thank you! The literature section is improved and a short discussion is added.

  1. Reviewer: It is also expected that the theoretical review will establish more relationships with LiDAR.

Authors: Thank you! A short discussion is added.

  1. Reviewer: It is not understood what "Material" is in the Material and Methods section. This phrase can be removed from the title.

Authors: Thank you! Corrected.

  1. Reviewer: Discussion section should also be improved.

Authors: Thank you! The manuscript structure is rearranged, so the Discussion section is expanded.

  1. Reviewer: In the Conlusions section, suggestions for future studies on how the theoretical framework can be applied in the practical field should be added.

Authors: Thank you! This is improved. Besides, English is improved throughout the paper.

Reviewer 2 Report

In this paper, the author presented the development of a theoretical model to study the two-photon excited fluorescence and plasma emission inside an aqueous dyed spherical (uranine-dyed water) droplet under the exposure of high-power laser radiation. Considering random distribution of nonlinear polarization sources in the droplet volume, the author simulated the angular far-field emission pattern and analysed the dependence of emission angular directivity on both the particle size and the type of optical nonlinearity.

From my point of view, the topic of this work is very good and is highly relevant to the concerned field of research. The scientific content of the manuscript and the presentation of the results in the figures is also very good. I, therefore, recommend the acceptance of the manuscript in its present form.

Author Response

We appreciate the time and effort of the reviewer for careful inspection of our work and good paper evaluation.

Reviewer 3 Report

A complete set of simulations of the scattering of a femtosecond laser beam penetrating in droplets of various sizes. This presentation will be useful for Lidar experiments. The paper is clear and easy to read.

Author Response

(The authors gave the same response as above.)

Reviewer 4 Report

The paper is good as it is and I do not have any further comments to make. I recommend it for publication.

Author Response

(The authors gave the same response as above.)

Reviewer 5 Report

1.       The main question addressed by the research is the development and dynamics of two-photon excited fluorescence and plasma emission inside an aqueous dyed spherical droplet is considered theoretically in particles exposed to a high-power laser radiation

2.       the topic original or relevant in the field

3.       What is LiDAR techniques? Please give some information about lidar  (Özdemir, S. , Akbulut, Z. , Karslı, F. & Acar, H. (2021). Automatic extraction of trees by using multiple return properties of the lidar point cloud . International Journal of Engineering and Geosciences , 6 (1) , 20-26 . DOI: 10.26833/ijeg.668352)

4.       What is mie scattering ? and what is the importance  at your study?

5.       Why did you use finite element method? Please mention about it and other methods

6.       Please add electromagnetic spectruma at your methods seçtin . Please explain it with figüre

7.       Please  check some referances

a)       Nazari, S. W., Akarsu, V., & Yakar, M. (2023). Analysis of 3D Laser Scanning Data of Farabi Mosque Using Various Softwaren . Advanced LiDAR3(1), 22–34. Retrieved from https://publish.mersin.edu.tr/index.php/lidar/article/view/975

b)      Çetin, Z. & Yastıklı, N. (2023). Automatic detection of single street trees from airborne LiDAR data based on point segmentation methods . International Journal of Engineering and Geosciences , 8 (2) , 129-137 . DOI: 10.26833/ijeg.1079210

1.       The main question addressed by the research is the development and dynamics of two-photon excited fluorescence and plasma emission inside an aqueous dyed spherical droplet is considered theoretically in particles exposed to a high-power laser radiation

2.       the topic original or relevant in the field

3.       What is LiDAR techniques? Please give some information about lidar  (Özdemir, S. , Akbulut, Z. , Karslı, F. & Acar, H. (2021). Automatic extraction of trees by using multiple return properties of the lidar point cloud . International Journal of Engineering and Geosciences , 6 (1) , 20-26 . DOI: 10.26833/ijeg.668352)

4.       What is mie scattering ? and what is the importance  at your study?

5.       Why did you use finite element method? Please mention about it and other methods

6.       Please add electromagnetic spectruma at your methods seçtin . Please explain it with figüre

7.       Please  check some referances

a)       Nazari, S. W., Akarsu, V., & Yakar, M. (2023). Analysis of 3D Laser Scanning Data of Farabi Mosque Using Various Softwaren . Advanced LiDAR3(1), 22–34. Retrieved from https://publish.mersin.edu.tr/index.php/lidar/article/view/975

b)      Çetin, Z. & Yastıklı, N. (2023). Automatic detection of single street trees from airborne LiDAR data based on point segmentation methods . International Journal of Engineering and Geosciences , 8 (2) , 129-137 . DOI: 10.26833/ijeg.1079210

Author Response

Respond to Reviewer#5

We appreciate the time and effort of the reviewer for careful inspection of our work.

N.B. All the revisions are high lightened in red.

  1. Reviewer: What is LiDAR techniques? Please give some information about lidar (Özdemir, S. , Akbulut, Z. , Karslı, F. & Acar, H. (2021). Automatic extraction of trees by using multiple return properties of the lidar point cloud . International Journal of Engineering and Geosciences , 6 (1) , 20-26 . DOI: 10.26833/ijeg.668352).

Authors: Thank you for the comment! The acronym LiDAR - Light Identification Detection and Ranging) - is now explained in the abstract. Besides, we have included the work suggested by the reviewer in the references list.

  1. Reviewer: What is mie scattering ? and what is the importance at your study?.

Authors: As commonly known, the term “Mie scattering” relates to the elastic light scattering on the spherical particles with the sizes less or comparable to the optical wavelength. Mie scattering stems from the works of German physicist G. Mie who along with L. Lorenz has elaborated the analytical solution to the Helmholtz equation in the form of infinite “Mie series”. We exploit the Mie scattering as a reference for the illustration of completely different nature of stimulated droplet emission.

  1. Reviewer: Why did you use finite element method? Please mention about it and other methods.

 Authors: Actually, there is no difference which numerical technique is used for such a simulation. One may use FDTD, FVM or FEM formulations. We choose FEM because of timely availability of COMSOL Multiphysics software which is based on this technique.

  1. Reviewer: Please add electromagnetic spectra at your methods section. Please explain it with figure.

 Authors: Worthwhile noting, as an important assumption in our simulations, we consider only a monochromatic optical radiation for both the excitation optical field and the nonlinear emission from the droplets. Thus, the only spectral information needed is the optical wavelengths of incident and emission radiation which now are explicitly indicated in the manuscript text and the caption to fig.1.

  1. Reviewer: Please check some references...

 Authors: Thanks for this proposal. However, after inspecting these papers we did not see they match the context of our work because they relate the software processing of LiDAR data obtained elsewhere. In our study, we do not discuss this point and focus on the physical model of laser-matter interaction which can be viewed as the source for LiDAR information.

Round 2

Reviewer 1 Report

Thank you for the revisions. I recommend that the manuscript be accepted for publication as it is.